# Association between Shift Work and Metabolic Syndrome: A 4-Year Retrospective Cohort Study

**DOI:** 10.3390/healthcare11060802

**Published:** 2023-03-09

**Authors:** Byeong-Jin Ye

**Affiliations:** Department of Occupational and Environmental Medicine & Institute of Environmental and Occupational Medicine, Busan Paik Hospital, Inje University, Busan 47392, Republic of Korea; yebj_oem@paik.ac.kr; Tel.: +82-51-890-6142; Fax: +82-51-893-3523

**Keywords:** shift work, metabolic syndrome, retrospective cohort study, generalized estimating equations

## Abstract

(1) Background: Previous studies on the association between shift work and metabolic syndrome have had inconsistent results. This may be due to the cross-sectional study design and non-objective data used in those studies. Hence, this study aimed to identify risk factors for Metabolic syndrome using objective information provided by the relevant companies and longitudinal data provided in health examinations. (2) Methods: In total, 1211 male workers of three manufacturing companies, including shift workers, were surveyed annually for 4 years. Data on age, smoking, drinking, physical activity, length of shift work, type of shift, past history, waist circumference, blood pressure, blood sugar, triglyceride, and high-density cholesterol (HDL) were collected and analyzed using generalized estimating equations (GEE) to identify the risk factors for Metabolic syndrome. (3) Results: In the multivariate analysis of Metabolic syndrome risk factors, age (OR = 1.078, 95% CI: 1.045–1.112), current smoking (OR = 1.428, 95% CI: 1.815–5.325), and BMI (OR = 1.498, 95% CI: 1.338–1.676) were statistically significant for day workers (n= 510). Additionally, for shift workers (N = 701), age (OR = 1.064, 95% CI: 1.008–1.174), current smoking (OR = 2.092, 95% CI: 1.854–8.439), BMI (OR = 1.471, 95% CI: 1.253–1.727) and length of shift work (OR = 1.115, 95% CI: 1.010-1.320) were statistically significant. Shift work was associated with a higher risk of Metabolic syndrome (OR = 1.093, 95% CI: 1.137–2.233) compared to day workers. For shift workers, shift work for more than 20 years was associated with Metabolic syndrome (OR = 2.080, 95% CI: 1.911–9.103), but the dose–response relationship was not statistically significant. (4) Conclusions: This study revealed that age, current smoking, BMI, and shift work are potential risk factors for Metabolic syndrome. In particular, the length of shift work (>20 years) is a potential risk factor for Metabolic syndrome in shift workers. To prevent metabolic syndrome in shift workers, health managers need to actively accommodate shift workers (especially those who have worked for more than 20 years), current smokers, and obese people. A long-term cohort study based on objective data is needed to identify the chronic health impact and the risk factors of shift work.

## 1. Introduction

Metabolic syndrome (MetS) is a state in which several conditions, including hyperglycemia, hypertension, hyperlipidemia, and obesity, occur concurrently [1], and it is defined as a cluster of risk factors that can directly facilitate the onset of arteriosclerotic cardiovascular disease [2]. The prevalence of Metabolic syndrome is rising steadily in developed and developing countries [3,4,5]. A meta-analysis revealed that Metabolic syndrome is associated with an elevated risk of coronary artery disease, myocardial infarction, and stroke [6,7]. Risk factors for Metabolic syndrome include aging [8], biological sex [9], body weight [10], low level of physical activity [11], alcohol consumption [12], smoking [13], and high-calorie intake [14].

The International Labour Organization defines shift work as a “method of organization of working time in which workers succeed one another at the workplace” during normal hours (9:00–18:00) or other hours and night hours [15]. The percentage of shift workers in the European Union rose from 17% in 2010 to 21% in 2015 [16], and in Korea, 9.7% of the working population were shift workers in 2017, with a continuous increase in the percentage over time [17]. 

Other studies on the association between shift work and Metabolic syndrome do not always show a positive association between the two factors [18,19,20,21,22,23]. The discrepancy in these results is likely due to cross-sectional design and non-objective data. Most previous studies have adopted cross-sectional designs. In a recent systematic literature review, the number of cross-sectional studies among articles meeting the researcher’s criteria was found to be much higher than cohort studies [24,25,26]. Previous studies have used subjective data from questionnaires to categorize work schedules crudely into shift and non-shift workers [27,28,29]; for example, a recent retrospective cohort study analyzed the association between Metabolic syndrome and shift work types using time series data [30] and another study investigated the association between the number of shifts and Metabolic syndrome components by calculating the number of shifts using daily work hours data. In light of the results [31], it is important to address the shortcomings of cross-sectional studies and non-objective data.

Therefore, this study aimed to identify the relationship between shift work and Metabolic syndrome using four-year health examination data of field workers, including shift workers working in manufacturing companies, and information related to shift work provided by the companies.

## 2. Materials and Methods

### 2.1. Study Design and Participants

This study was conducted using the health examination data of male workers (including shift workers) in three manufacturing companies. A total of 1438 workers who completed an annual health examination for 4 years from 2015–2018 were selected. From this population, workers with missing data on some essential tests required for the diagnosis of Metabolic syndrome (n = 28), workers diagnosed with Metabolic syndrome in 2015 (n = 158), and workers who experienced a change in their shift work schedule between 2015 and 2018 (n = 41) were excluded, resulting in a total of 1211 participants being selected, of whom 701 were shift workers, and 510 were day workers (Figure 1).

At each health examination, a physician questioned workers about their work department, length of employment, shift work status, and changes in shift schedules. Shift workers were required to complete a night-shift questionnaire during their health checkups. This questionnaire contained questions about the type of shift work and the length of shifts.

### 2.2. Shift Work Status

In this study, shift work and shift workers were defined by the regulations of the Korean Ministry of Employment and Labor [30].

Night shift: Work performed between 22:00 and 6:00.Night-shift workers: Workers who met any one of the following criteria:
Worked at least 60 h of night shifts per month for 6 months;Worked an average of four shifts between 22:00 and 6:00 for six consecutive months.


Shift work was categorized into rotating 8 h shifts and rotating 12 h shifts. Rotating 8 h shifts were divided into morning, evening, and night shifts. Rotating 12 h shifts were divided into day shifts and night shifts. The companies participating in the study consisted of rotating 8 h and rotating 12 h shifts.

The worker was divided into daily work and shift work according to their department and job. The shift work consisted of two shifts and three shifts. However, changes in the type of shift work were often experienced according to the circumstances of the company. The duration of the shift was verified by the company’s health manager through the employee’s work history based on the employee’s personnel information data. Therefore, the duration of the shift is measured from the time of joining the company to the period of study. The type of shift work was confirmed through health checkup questionnaires and doctor-led interviews but was limited to the study period due to complex shift work pattern changes and memory dependence.

### 2.3. Assessment of Metabolic Syndrome

In this study, Metabolic syndrome was diagnosed based on the modified National Cholesterol Education Program (NCEP) Adult Treatment Panel (ATP) III criteria [32]. Metabolic syndrome were diagnosed if at least three of the following five criteria were met:Blood pressure (BP) ≥ 130/85 mmHg or currently undergoing treatment for hypertension;Fasting glucose ≥ 100 mg/dL or currently undergoing treatment for elevated blood sugar;High serum triglyceride (≥150 mg/dL) or currently undergoing treatment for dyslipidemia;Low serum high-density lipoprotein (HDL): <40 mg/dL for males, <50 mg/dL for females, or currently undergoing treatment for dyslipidemia;Waist circumference (WC) ≥90 cm for males, ≥80 cm for females.

### 2.4. Data Collection and Measures

Past history, smoking, drinking, physical activity, length of shift work, and type of shift work were recorded for the period from 2015 to 2018 using annual history-taking and self-reported questionnaires. Waist circumstance, obesity, blood pressure, blood glucose, triglyceride, and HDL were determined with physical examination and blood tests.

Smoking status was categorized into current smoker and non-smoker. Drinking was divided into drinker (drinking alcohol at least once a week) and non-drinker (drinking alcohol less than once a week). Physical activity was defined as aerobic activity and measured using the Korean version of the short form of the International Physical Activity Questionnaire [33]. Physical activity was divided into adequate physical activity and inadequate physical activity, based on the criteria of moderate-intensity exercise for ≥2.5 h per week or high-intensity exercise for ≥1.25 h per week (one minute of high-intensity exercise = two minutes of moderate-intensity exercise). The duration of the shift was verified by the company’s health manager through the employee’s work history based on the employee’s personnel information data. Thus, the length of shift work was defined as the number of years of shift work since hiring, and the type of shift work was confirmed through health checkup questionnaires and doctor interviews. However, the type of shift was determined based on the type of shift work performed during the study period due to complex shift work pattern changes and memory dependence.

### 2.5. Statistical Analysis

First, we compared age, drinking status, smoking status, and physical activity in 2015 and differences in the prevalence of Metabolic syndrome in 2015 and 2018 between day workers and shift workers using t-tests and chi-square tests. The length and type of shift among shift workers were analyzed according to the mean (standard deviation) and frequencies.

We used generalized estimating equations (GEE) to analyze the risk factors for Metabolic syndrome based on the four years of annual health examination data for each individual worker. The risk factors of Metabolic syndrome in the day workers and shift workers were analyzed by performing multivariate analyses. The factors used in the analyses are the well-known risk factors for Metabolic syndrome, namely age, smoking, drinking, physical activity obesity, and shift work. Length and type of shift work were also used in the analysis related to shift workers. Then, we analyzed the effects of shift work and shift hours, which are statistically significant factors in multivariate analysis, on Metabolic syndrome. The data were analyzed using SPSS software (Version 25.0; IBM Corp., Armonk, NY, USA).

## 3. Results

In 2015, the first year of the study, age (*p* = 0.135), drinking (*p* = 0.588), BMI (*p* = 0.784), and physical activity (*p* = 0.319) did not differ between day workers and shift workers. Current smoking (*p* < 0.001) differed significantly between day workers and shift workers. Among shift workers, the mean length of shift work was 16.27 + 9.11 years and there were more workers in three shifts (87.75%) than in two shifts (12.25%). The prevalence of Metabolic syndrome in 2015 and 2018 was statistically different between day workers and shift workers (*p* < 0.001, *p* = 0.028), respectively (Table 1). The incidence rates of metabolic syndrome in day workers and night-shift workers were 5.3 and 10.5 cases per 1000 person-years, respectively.

As a result of the multivariate analysis of Metabolic syndrome risk factors, age (OR = 1.069, 95% CI: 1.049–1.090), current smoking (OR = 2.006, 95% CI: 1.428–2.817) and BMI (OR = 1.378, 95% CI: 1.294–1.467) were found to be statistically significant for all workers. Age (OR = 1.078, 95% CI: 1.045–1.112), current smoking (OR = 1.428, 95% CI: 1.815–5.325), and BMI (OR = 1.498, 95% CI: 1.338–1.676) were statistically significant for day workers. Furthermore, for shift workers, age (OR = 1.064, 95% CI: 1.008–1.174), current smoking (OR = 2.092, 95% CI: 1.854–8.439), BMI (OR = 1.471, 95% CI: 1.253–1.727), and length of shift work (OR = 1.155, 95% CI: 1.010–1.320) were statistically significant (Table 2).

In the multivariate analysis of the effect of shift work on Metabolic syndrome, shift work was positively associated with Metabolic syndrome in all models (Model 1 (OR = 1.204, 95% CI: 1.080–1.479), Model 2 (OR = 1.444, 95% CI: 1.164–1.791), Model 3 (OR = 1.093, 95% CI: 1.137–2.233) (Table 3)).

In the multivariate analysis performed by dividing the three groups by length of shift work, there was a positive correlation with Metabolic syndrome in the group that worked shifts for more than 20 years in all models (model 1 (OR = 2.227, 95% CI: 1.469–3.376), Model 2 (OR = 2.847, 95% CI: 1.500–5.403), Model 3 (OR = 2.080, 95% CI: 1.911–9.103)). The dose–response relationship was statistically significant in Model 2, but not significant in Model 1 and Model 3 (Table 4).

## 4. Discussion

This study retrospectively reviewed the 4-year annual health examination data for the field workers of manufacturing companies. In addition, the work history information obtained through the company’s personnel data was used to confirm the shift work status and shift work period and results of recurrent annual tests for the same workers; we also ensured that data were adjusted for intra- and inter-personal changes, as was the case for the panel data.

The incidence rates of metabolic syndrome in day workers and night shift workers were 5.3 and 10.5 cases per 1000 person-years, respectively, in this study. This is a considerable difference compared to the prevalence (25.6% in men and 12.4% in women) in the analyzed data obtained from the National Health and Nutrition Examination Survey conducted from 2016 to 2018 in Korea [34]. A possible explanation for this is the difference in age distribution and activity level. The proportion of people aged 60 and older with a high prevalence of metabolic syndrome is very low in this study (0.5% vs 16.4%). Additionally, although there was a difference in terms of the measurement method (physical activity VS sitting time), the normal sitting time rate was found to be 42.9%. However, on the other hand, the appropriate physical activity rate (91.77% for day workers and 88.24% for shift workers) in this study was found to be very high.

In this study, shift work was a significant risk factor for Metabolic syndrome, which is similar to previous findings [35,36]. Previous studies have suggested that circadian rhythm disturbance may induce Metabolic syndrome [37,38,39]. A number of previous studies demonstrated that shift work does not significantly increase metabolic syndrome risk. A study of 3008 shift workers and 8015 day workers, specifically male railway workers, who worked for more than 10 years found no link between shift work and metabolic syndrome [40]. However, since the age was limited to 40 years of age or older, it is possible that the development of metabolic syndrome in young workers who work shifts was overlooked. In addition, in a study of 3007 male employees (1700 day and 1307 shift workers) at a car-manufacturing company, two-shift work was associated with a lower risk of metabolic syndrome than day work [18]. However, this study had limitations in that it was a cross-sectional study in which the status of shift work in the past was unknown and that it did not consider differences in work according to department. The present study included young workers as well as older workers, and objectively identified the past shift work status. Additionally, since field workers were included in the study, the difference in work by the department may not be large. The difference between these study participants and methods might help to explain the difference in results. 

Furthermore, our findings showed elevated ORs for Metabolic syndrome after 20 years of shift work. Similar findings have been reported previously. A meta-analysis of 13 observational studies published between 1971 and 2013 [41] confirmed a dose–response relationship between the length of shift and Metabolic syndrome. A cross-sectional study of 134 male blue-collar workers also reported that the risk for Metabolic syndrome increased with >30 years of shift work [42]. Considering that examining the onset of chronic disease, such as Metabolic syndrome, as an outcome in relation to shift work requires the study of a specified time period, it appears that the length of shift work [43] and the frequency of shift work [23] are more meaningful determinants than working shifts. However, in this study, the dose–response relationship could not be confirmed. Further research using longitudinal data should be carried out in the future.

Obesity is a known risk factor for Metabolic syndrome, and this was also observed in this study. The results of this study were consistent with previous reports that BMI is related to WC, an Metabolic syndrome component, and shift work is positively correlated with obesity [44,45]. Smoking was significantly associated with Metabolic syndrome. It has been reported that the risk of Metabolic syndrome increases with increasing cigarette consumption [12,13]. Although drinking is also a lifestyle factor linked to the onset of Metabolic syndrome [46,47], no significant differences were observed in this study. Changes in drinking frequency and drinking status should be further examined in a larger study population. Age is a known risk factor for Metabolic syndrome [48,49], as confirmed in this study; this was consistent with the findings of previous studies. In addition, several studies also report no difference in physical activity between day and shift workers [50,51], which is consistent with our findings.

In this study, a significant association between the type of shift work and Metabolic syndrome was not observed. Previous study findings on Metabolic syndrome and rotating 12 h shifts or 8 h shifts are inconsistent [52,53]. It is possible that the observation period of shift work patterns in this study was too short (4 years) and the changes in shift work patterns were too complex, which led to a scarcity of data in the analysis. Considering that two-shift work is more likely to cause circadian rhythm disturbances than three-shift work, it is necessary to extend the observation period in future studies.

### 4.1. Limitations

This study has a few limitations. First, the study period was rather short (4 years). The Korean government launched a worker’s health examination program for shift workers in 2014 [2], and the program is still in effect. However, prior to this program, collecting data on factors such as length of shift work and type of shift work was difficult. Longer follow-up studies of shift work should be conducted to investigate the health impacts. Second, we did not have information on workers’ education level and dietary patterns, so we were unable to adjust for these variables as potential confounders in the multivariate analyses. Future studies should include these variables to clarify the metabolic risk mechanism affecting shift workers. Third, molecular-based studies are important for identifying any contributing mechanisms. However, this study was based on health screening, meaning that adiponectin, leptin, other inflammatory markers, or oxidative stress markers were not included in this study. Fourth, the use of alcohol units for drinking and pack-years for smoking are ways to further increase the objectivity of the study. However, due to a lack of data, alcohol units, and pack-years were not used in this study. Since the lack of objectivity of the variables can cause confusion in the interpretation of the results and limit comparisons with other studies, future study designs should aim to further increase the objectivity of the variables. Finally, we could not collect data on sleep conditions due to missing data. Sleep deprivation is linked to circadian misalignment in shift workers [54], and circadian misalignment has been identified as a fundamental cause of poor cardiovascular health [55]. Future studies should include sleep status to clarify mechanisms affecting cardiovascular diseases.

### 4.2. Strengths

Despite these limitations, this study used a 4-year longitudinal dataset obtained via annual health examinations for the same population, similar to panel data, and a time series analysis was performed using GEE. Moreover, the classification of shift workers and day workers and shift work period was performed according to the work histories available in the company’s personnel records. Hence, there is little risk of errors in participant classification and shift work duration.

## 5. Conclusions

In conclusion, this study investigated the same workers over a 4-year period using annual data. In particular, by using objective data, the risk of errors in shift work classification and shift work period was lowered. The study revealed that shift work, the length of shift work, age, obesity, and smoking are potential risk factors for Metabolic syndrome. To prevent metabolic syndrome in shift workers, health managers need to actively take care of shift workers (especially those who worked for more than 20 years), current smokers, and obese people. Longer studies and time series analyses are required to shed light on the chronic health impact of shift work and the mechanisms underlying these risk factors.

## Figures and Tables

**Figure 1 healthcare-11-00802-f001:**
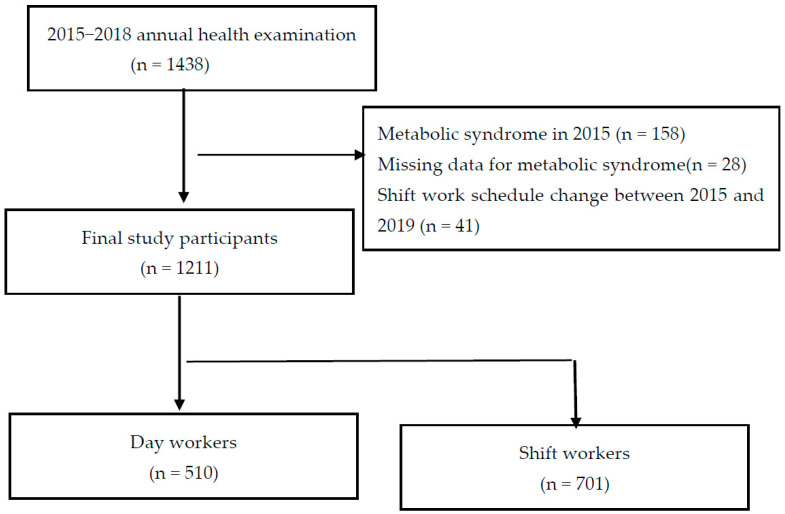
Flowchart of the study participants’ selection.

**Table 1 healthcare-11-00802-t001:** Demographic, lifestyle, and occupational characteristics of the study participants in 2015 and Metabolic syndrome at 2015 and in 2018 for day workers and shift workers.

		Day Workers n (%)	Shift Workers n (%)	*p*-Value ^1^
Age(yr) ^2^ (mean ± SD)		44.04 ± 9.35	43.26 ± 8.401	0.135
Current smoking	Yes	216 (43.11)	383 (55.42)	<0.001
	No	285 (56.89)	308 (44.58)	
Drinking	Drinker	143 (62.72)	178 (60.34)	0.588
	Non-drinker	85 (37.28)	117 (39.66)	
Physical activity	Inadequate	20 (8.23)	33 (11.07)	0.310
	Adequate	223 (91.77)	265 (88.24)	
BMI (kg/m^2^) ^2^ (mean ± SD)		24.03 ± 2.93	24.08 ± 3.12	0.784
Length of shift work (yr) ^2^ (mean ± SD)			16.27 ± 9.11	
Type of shift work	2 shifts		37 (12.25)	
	3 shifts		265 (87.75)	
MetS in 2015	Yes	42 (7.46)	116 (14.39)	<0.001
	No	521 (92.54)	690 (85.61)	
MetS in 2018	Yes	16 (3.12)	42 (5.99)	0.028
	No	494 (96.88)	659 (94.01)	

Values for categorical variables given as a number (percentage); values for continuous variables given as mean ± SD. ^1^
*χ*_2_ test, categorical variables; *t*-test, continuous variables. ^2^ Continuous variables.

**Table 2 healthcare-11-00802-t002:** Odds ratio (OR) and 95% confidence interval (CI) of risk factors for MetS in all workers, day workers, and shift workers.

	All Workers		Day Workers		Shift Workers	
	OR (95% CI)	*p*-Value	OR (95% CI)	*p*-Value	OR (95% CI)	*p*-Value
Age (yr)	1.069 (1.049–1.090)	≤0.001	1.078 (1.045–1.112)	≤0.001	1.064 (1.008–1.174)	0.004
Current smoking (Reference: no)	2.006 (1.428–2.817)	≤0.001	1.428 (1.815–5.325)	≤0.001	2.092 (1.854–8.439)	0.001
Drinking (Reference: non-drinker)	1.298 (0.918–1.835)	0.140	0.812 (0.482–1.370)	0.435	1.526 (0.541–4.305)	0.425
Physical activity (Reference: adequate)	1.131 (0.820–1.560)	0.454	1.484 (0.901–2.444)	0.121	0.581 (0.221–1.527)	0.271
BMI (kg/m^2^)	1.378 (1.294–1.467)	≤0.001	1.498 (1.338–1.676)	≤0.001	1.471 (1.253–1.727)	≤0.001
Length of shift work (yr)					1.155 (1.010–1.320)	0.035
Type of shift (Reference: 3 shift)					1.986 (0.847–4.147)	0.951

ORs were calculated using generalized estimating equations (GEE). All workers and day workers: adjusted for age, BMI, current smoking, drinking, and physical activity. Shift workers: adjusted for age, BMI, current smoking, drinking, physical activity, length of shift work, and type of shift.

**Table 3 healthcare-11-00802-t003:** Odds ratio (OR) and 95% confidence interval (CI) of MetS by the shift work.

	Model 1		Model 2		Model 3	
	OR	95% CI	OR	95% CI	OR	95% CI
Day work	1		1		1	
Shift work	1.204	1.080–1.479	1.444	1.164–1.791	1.093	1.137–2.233

ORs were calculated using generalized estimating equations (GEE); Model 1: single factor (shift work); Model 2: adjusted for age and BMI; Model 3 adjusted for age, BMI, smoking, drinking, and physical activity.

**Table 4 healthcare-11-00802-t004:** Odds ratio (OR) and 95% confidence interval (CI) of MetS according to the length of shift work in shift workers.

	Model 1		Model 2		Model 3	
	OR	95% CI	OR	95% CI	OR	95% CI
1–9	1		1		1	
10–19	1.699	0.969–2.699	2.152	1.249–3.706	2.178	0.783–6.056
≥20	2.227	1.469–3.376	2.847	1.500–5.403	2.080	1.911–9.103

ORs were calculated using generalized estimating equations (GEE); Model 1: single factor (shift work duration). Model 2: adjusted for age, and BMI; Model 3 adjusted for age, BMI, smoking, drinking, physical activity, and type of shift.

## Data Availability

The data presented in this study are available on request from the corresponding author. The data are not publicly available due to privacy.

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
