# Peer review of "Association between Shift Work and Metabolic Syndrome: A 4-Year Retrospective Cohort Study"

_healthcare, 2023, doi:10.3390/healthcare11060802_

Round 1

Reviewer 1 Report

Physical activity is not a risk factor but physical inactivity or low level of physical activity. Please correct this in your introduction. Likewise dietary habits.

Please specify reference categories for all variables. Currently, you specified only gender.

Writing ‘reference’ to indicate reference categories in Tables 3 and 4 is redundant since you have mentioned that category and also indicated 1 as the OR.

For confidence interval, use en-dash to indicate the range.

In Table 2, risk factors are similar for both groups of workers. I suggest you have a total column to investigate these risk factors for all the workers.

You had only 1.9% and 1.1% female workers in the groups. Don’t you think this adversely affected your comparison? I would drop gender from the analysis because of the inadequate representation of the female gender.

In Table 1, specify n and %

Since you have data, I would love to see the differences in the incidence of MetS among the day and shift workers.

Please check punctuation and grammar. I noticed a few errors. For example, in line 209, there is a full stop in an inappropriate place.

In the abstract, you mentioned, previous studies on the association between shift work and metabolic syndrome (MetS) have had inconsistent results. However, in the discussion, you supported your findings with only the studies which presented similar findings. I expect, you also cite the studies which presented different findings and discuss the possible reasons for the differences between your study and those.

You recommended that to prevent metabolic syndrome in shift workers, Health managers should actively encourage shift workers for more than 20 years to quit smoking and lose weight. Did you investigate the interaction between these risk factors and shift work? Your result suggests an association between shift work and MetS after adjusting for the confounding effect of smoking and obesity! I would like to know what is the basis of your recommendation.

Author Response

Thank you for your comments and suggestions.

Reviewer 2 Report

- Be careful with minor grammatical or structural mistakes in writing. e.g. 

line 211 - In this study, Shiftwork...    the work shiftwork should be small capital.

Line 209- period. and results....         there should not be full stop after the word period.

Author Response

(The authors gave the same response as above.)

Reviewer 3 Report

The aim was to assess the association between shift work and metabolic syndrome in a 3-year follow-up study. The data are in line with these of a meta-analysis. The pooled OR of metabolic syndrome in shift workers based on 12 studies was 2.17 (95% CI = 1.31-3.60, P = 0.003; I2 = 82%, P < 0.001). (Obes Rev 2022 Oct;23(10):e13489).

However, this study raises some significant concerns.

1)     Unlike in other studies, there was no correlation between the length of shift work and the risk of metabolic syndrome. However, the mean length of shift work was 16.61+9.22 166 years; thus very long compared to the follow-up period.

2)      A study published in Scientific Reports (2022) reported that the metabolic syndrome prevalence in healthy Korean adults was 25.6% and 12.4% in men and women, respectively. So why is the prevalence of metabolic syndrome much lower than expected, even among shift workers?

3)      The study should be confined to men because the number of women is too small.

1)      Previous studies showed that sleep quality, longer sleep latency, shorter sleep duration, and sleep disturbances strongly correlated with the risk of metabolic syndrome. However, this does not account for differences in sleep. Therefore, it is interesting to look at differences between workers in 3 and 2 shifts.

4)      The author stated that most studies, unlike his, lacked objective criteria. However, workers were divided into drinkers (alcohol drinking at least once a week) and non-drinkers (alcohol drinking less than once a week). However, it would be better to include units of alcohol. For smokers, it would be better to consider pack years.

5)      As in many other studies, no molecular basis was determined. Were data on adiponectin, leptin, other inflammatory markers, or markers of oxidative stress available?

Minor comment: Table 1 should indicate the number of persons with metabolic syndrome at the start, showing a relatively small increase over three years.

Author Response

(The authors gave the same response as above.)

Round 2

Reviewer 1 Report

Thanks for addressing my comments!

Author Response

Thank you for your interest
I revised the English language and style with the help of the MDPI English editing system.

Reviewer 3 Report

The authors responded to some of my concerns. 

However, this is not a follow-up study. The authors did not compare the increase in metabolic syndrome between shift and non-shift workers over 4 years.

The authors did not use objective criteria for drinking, smoking and sleeping. 

Therefore, the authors cannot claim in the abstract that their study was better than previous studies. In fact these major shortcomings should be mentioned in the discussion.

Author Response

Thank you for your attention and detailed comments.

I agree that this study is not a follow-up study. therefore, I remove the word "follow-up " from the text. In addition, the content on the lack of objectivity on drinking and smoking was inserted into the discussion as follows.

Fourth, the use of alcohol units for drinking and pack-years for smoking are ways to further increase the objectivity of the study. However, due to a lack of data, alcohol units, and pack-years were not used in this study. Since the lack of objectivity of the variables can cause confusion in the interpretation of the results and limit comparisons with other studies, future study designs should aim to further increase the objectivity of the variables.

finally, I revised the English language and style with the help of the MDPI English editing system.
